# Application of Improved Particle Swarm Optimisation Algorithm in Hull form Optimisation

**Qiang Zheng [1,2], Bai-Wei Feng [1,2,*], Zu-Yuan Liu [1,2,*] and Hai-Chao Chang [1,2]**

1  Key Laboratory of High Performance Ship Technology, Wuhan University of Technology, Ministry of Education, Wuhan 430063, China; 226207@whut.edu.cn (Q.Z.); changhaichao@whut.edu.cn (H.-C.C.)
2  School of Naval Architecture, Ocean and Energy Power Engineering, Wuhan University of Technology, Wuhan 430063, China
*  Correspondence: fengbaiwei@126.com (B.-W.F.); wtulzy@whut.edu.cn (Z.-Y.L.)

**Abstract:** The particle swarm optimisation (PSO) algorithm has been widely used in hull form optimisation owing to its feasibility and fast convergence. However, similar to other intelligent algorithms, PSO also has the disadvantages of local premature convergence and low convergence performance. Moreover, optimization data are not used to analyse and reduce the range of values for relevant design variables. Our study aimed to solve these existing problems in the PSO algorithm and improve PSO from four aspects, namely data processing of particle swarm population initialisation, data processing of iterative optimisation, particle velocity adjustment, and particle cross-boundary configuration, in combination with space reduction technology. The improved PSO algorithm was used to optimise the hull form of an engineering vessel at $Fn$ = 0.24 to reduce the wave-making resistance coefficient under static constraints. The results showed that the improved PSO algorithm could effectively improve the optimisation efficiency and reliability of PSO and effectively overcome the drawbacks of the PSO algorithm.

**Keywords:** particle swarm optimisation (PSO); space reduction; hull form optimisation; wave-making resistance coefficient

## 1. Introduction

With the continuous improvement of computer processing power and the accuracy of computational fluid dynamics (CFD), CFD-based hull form optimisation has been rapidly developed. This directly applies hull surface parametric modification, CFD numerical simulations, and optimisation technologies to the design of new ships and eventually obtains the hull form with the best performance under given constraints. Many scholars have carried out considerable research on hull form optimisation, including D. Peri [1,2] of Italian Ship Model Basin, Rome; S. Harries [3] of the Technische Universität Berlin; Yang C [4–6] of George Mason University, USA; and Feng Baiwei [7–9] and Chang Haichao et al. [10–12] of Wuhan University of Technology.

At present, the research pertaining to hull form optimisation includes the following. (1) Design space reduction. Harries and Abt [13] reported a massive reduction of parameters that spanned the design space and utilized the gradient information as derived from adjoint simulations. D'Agostino et al. [14] and Serani et al. [15] reduced the dimensionality of the design space by providing a shape reparameterization using Karhunen–Loeve expansion/principal component analysis (KLE/PCA) eigenvalues and eigenmodes. Khan et al. [16] adopted a two-step learning methodology to identify a lower-dimensional latent space based on the combination of geometry- and physics-informed principal component analysis and active subspace method, which can be utilized for efficient design exploration and the construction of improved surrogate models for physics-based prediction of designs. Tezzele et al. [17] pioneered the application of a methodology based on active subspace properties to the naval architecture field for parameter space

reduction. (2) Metamodel. Serani et al. [18] reported a study on four adaptive sampling methods of a multi-fidelity metamodel, based on stochastic radial basis functions (RBFs), to achieve a global design optimisation using expensive CFD computer simulations and adaptive grid refinement. Chunna et al. [19] proposed an effective global optimisation method that integrates the adaptive filling algorithm based on fuzzy clustering into the kriging model. Zhang et al. [20] proposed a data prediction method based on improved particle swarm optimisation (IPSO)-Elman NN to improve the prediction accuracy of total resistance. Coppedè et al. [21] proposed a Gaussian process-response surface method (GP-RSM) based on an ordinary kriging model, which was developed to enhance the evaluation speed of the quantity of interest in the design process and applied to improve the calm-water performance of the KCS. (3) Efficient optimisation algorithm. Pellegrini et al. [22] proposed a multi-objective derivative-free and deterministic global/local hybrid algorithm as an efficient and effective solution to SBDO problems. Tezdogan et al. [23] proposed a hybrid algorithm to solve the complicated nonlinear optimisation problem of fishing boat. Serani et al. [24] presented study of DPSO, with application to simulation-based design in ship hydrodynamics. Leotardi [25] describes a class of novel initializations in deterministic particle swarm optimization (DPSO) for approximately solving costly unconstrained global optimization problems.

Owing to the complex spatial characteristics of hull form design optimisation, a general mathematical method cannot provide feasible solutions in the actual optimisation process. Therefore, intelligent optimisation algorithms, such as a genetic algorithm (GA) or particle swarm optimisation (PSO), are used in hull form optimisation to obtain approximately optimal solutions. PSO is a population-based stochastic optimisation technique introduced by Kennedy and Eberhart in 1995 [26,27]. The PSO algorithm is easier to implement, has fewer parameters, and has proven to converge faster than conventional optimisation methods such as GA [28,29]. Owing to its simplicity, ease of implementation, and high convergence speed [30,31], the PSO algorithm has been widely used for hull form optimisation.

Similar to other intelligent optimisation algorithms, the PSO algorithm also exhibits the disadvantages of local premature convergence and low convergence performance. To solve such problems, many scholars have improved PSO. Because the PSO algorithm falls into local optimisation in high-dimensional problems, Cheng et al. [30] modified the particle diversity in the optimisation process to improve the optimisation performance. Mathew M. Noel [31] combined PSO with the gradient algorithm to increase the local optimisation ability, thereby improving the algorithm optimisation efficiency and optimisation progress. Shi et al. [32] made a comprehensive improvement on the inertial weight adjustment strategy, differential evolution, and local variable depth search of PSO, which reasonably and efficiently balanced the global and local search abilities of the algorithm. Reungsinkonkarn et al. [33] applied search space reduction (SSR) to PSO to eliminate the optimal region that may not find the optimal solution through SSR and to improve the algorithm optimisation efficiency. Zhang et al. [34] proposed a multi-objective discrete PSO algorithm based on a fine perturbation strategy (EPSMODPSO), which performs well in the diversity and convergence of the obtained Pareto optimal frontier. It can reconfigure the ship power system and solve other multi-objective discrete optimisation problems. To improve the premature convergence and low search accuracy of conventional PSO, Wang et al. [35] proposed PSO with an enhanced global search and local search (EGLPSO) to improve global and local search. This algorithm can greatly improve the performance of conventional PSO in terms of search accuracy, search efficiency, and global optimality.

Our study achieved certain improvement of the PSO algorithm in terms of its existing problems. In the process of particle initialisation and optimisation iteration, the space reduction method was introduced to reduce the particle-changing space in the optimisation process. The particle velocity and the cross-boundary particles after space reduction were processed to improve the particle diversity in the optimisation process. The method was verified by function examples. The results showed that the improved algorithm

can improve the optimisation efficiency while ensuring the optimisation performance. Finally, the improved PSO algorithm was applied to the hull form optimisation of an engineering vessel.

This article has five sections. Section 2 introduces the space reduction technique. Section 3 describes the improved PSO algorithm. Section 4 verifies the feasibility of the improved optimisation algorithm by function examples. Section 5 applies this algorithm to the optimisation of the bow shape of a certain engineering ship. Section 6 summarises the research and describes the future directions.

## 2. Space Reduction Technique Based on Partial Correlation Analysis

Space reduction is a practical division and reduction of the design space through exploration and analysis, including dimensional reduction and size reduction. Hull form optimisation is a typical and complex engineering problem. It contains a large amount of numerical simulation with complex spatial design performance, leading to low optimisation efficiency and difficulty in obtaining global optimal solutions. To improve its efficiency and performance, many scholars have introduced space reduction into hull form optimisation [8,36]. In our study, we used the partial correlation analysis for design space reduction.

### 2.1. Partial Correlation Analysis

A correlation coefficient symmetry matrix composed of simple coefficients is required to calculate the partial correlation coefficient [37,38]. There are n variables $X_1$, $X_2$, ... , $X_n$, the Pearson correlation coefficient between any two variables $X_i$, $X_j$ is $r_{ij}$(i, j = 1, 2, ... , n), and the Pearson correlation coefficient is calculated as

$$r_{ij} = \frac{\sum (X_i - \overline{X}_i)(X_j - \overline{X}_j)}{\sqrt{\sum (X_i - \overline{X}_i)^2 (X_j - \overline{X}_j)^2}} \tag{1}$$

where $\overline{X}$ is the average. The following correlation coefficient symmetry matrix is composed of simple correlation coefficients:

$$R = \begin{bmatrix} r_{11} & r_{12} & \cdots & r_{1n} \\ r_{21} & r_{22} & \cdots & r_{2n} \\ \cdots & \cdots & \cdots & \cdots \\ r_{n1} & r_{n2} & \cdots & r_{nn} \end{bmatrix} \tag{2}$$

The partial correlation coefficient between any two variables $X_i$ and $X_j$ is $R_{ij}$, and the equation for calculating the partial correlation coefficient is

$$R_{ij} = \frac{-\Delta_{ij}}{\sqrt{\Delta_{ii} \cdot \Delta_{ii}}} \tag{3}$$

where $\Delta_{ij}$, $\Delta_{ii}$, and $\Delta_{jj}$ are algebraic cofactors corresponding to elements $r_{ij}$, $r_{ii}$, and $r_{jj}$ in in the symmetric matrix of the correlation coefficient, respectively.

The correlation coefficient between variables and objectives can be obtained with appropriate data samples using the above theory. A positive sign before the partial correlation coefficient represents a positive linear relationship between the variable and the target; the negative sign represents a negative linear relationship between the variable and the target; the distribution range of the coefficient is between 0 and 1; and the numerical value indicates the linear correlation between the design variable and the target. The feasibility study of partial correlation analysis for space reduction is detailed in Wu's article [36].

### 2.2. Partial Correlation Coefficient and Degree-of-Space Reduction

The relationship between the value of the partial correlation coefficient and the spatial distribution of variables is investigated by using a single objective function. Equation (4) is used in the first step.

$$y = |x| \tag{4}$$

When $x \in [0, 1]$, the relationship between the target and variable is as shown in Figure 1. The figure shows that there is a positive linear relationship between the target and the variable. When $x = 0$, the minimum value of the function $y = 0$ is obtained. The distribution space of x is divided into two equal parts; (0, 0.5) is defined as the lower space, (0.5, 1) is the upper space, and $x = 0.5$ is the median of the variable, which is the midpoint of the x distribution space. Fifty random samplings are performed to calculate the function values and perform partial correlation analysis. The obtained partial correlation coefficient between $y$ and $x$ is 1. It is concluded that the smaller the value of $x$, the is smaller the value of $y$. Therefore, the upper space of the $x$ variable is irrelevant for obtaining the minimum value of the function and can be discarded.

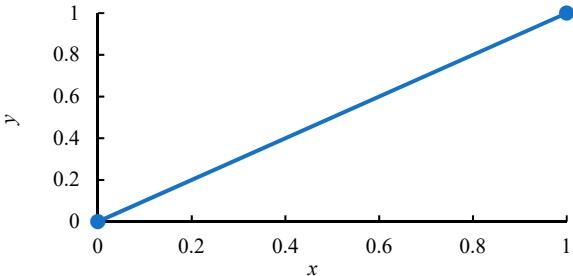

**Figure 1.** Function relationship plot ($x \in [0, 1]$).

When $x \in [-1, 1]$, the relationship between the target and the variable is as shown in Figure 2. It is impossible to analyse the correlation between the variable and the function value from the overall space of the variable. Similarly, the distribution space of $x$ is divided into two equal parts. When x is the variable median value $x = 0$, the function value is smallest. One hundred samplings are randomly to calculate the function values and perform partial correlation analysis. The obtained partial correlation coefficient between $y$ and $x$ is 0.0055. Theoretically, the partial correlation coefficient obtained by uniform sampling in the variable space should be 0. The resulting partial correlation coefficient is a value close to 0 owing to random sampling. Therefore, it is impossible to determine whether the upper and lower spaces are valuable for obtaining the minimum value of the function, and these spaces cannot be discarded.

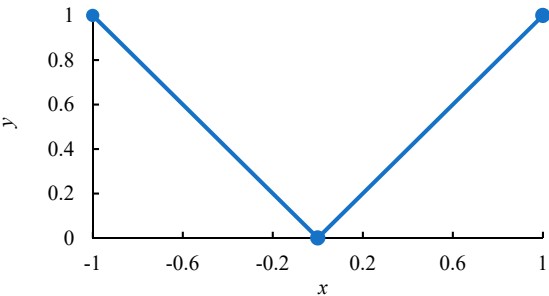

**Figure 2.** Function relationship plot ($x \in [-1, 1]$).

The function with the local maximum value (Equation (5)) is selected for further study. The relationship between the target and variable is as shown in Figure 3. The characteristic of this function is that there are many local optimal solutions, and the minimum value $y = 0$ is obtained when $x = 0$, which is closer to the actual optimisation than Equation (4).

$$y = \sin(x^2) + |x| \tag{5}$$

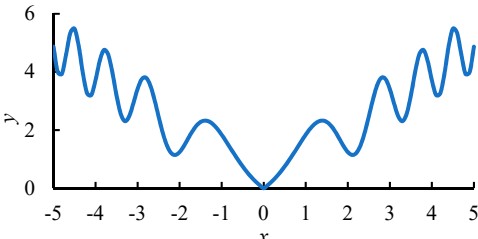

**Figure 3.** Function relationship plot.

The function is randomly sampled 1000 times in different variable ranges, and a partial correlation analysis is performed. The results are shown in Table 1. By testing the ranges of different x values, it can be found that when the optimal value is the median of the variable, the partial correlation coefficient is close to 0; when the optimal value is farther from the median of the variable and is closer to the boundary value of a side, the partial correlation coefficient is close to 1. Similarly, it can be concluded that when the partial correlation coefficient is 0, the space on both sides of the median value cannot be discarded; when the partial correlation coefficient is 1, the space on one side can be discarded by determining the sign of the partial correlation coefficient.

**Table 1.** Function analysis results.

| Range of $x$ | $[-5,5]$ | $[-4,5]$ | $[-3,5]$ | $[-2,5]$ | $[-1,5]$ | $[0,5]$ |
|---|---|---|---|---|---|---|
| Variable median | $x = 0$ | $x = 0.5$ | $x = 1$ | $x = 1.5$ | $x = 2$ | $x = 2.5$ |
| Optimal value | | | $x = 0$ | | | |
| Partial correlation coefficient | $-0.021$ | $0.260$ | $0.524$ | $0.736$ | $0.872$ | $0.894$ |

Therefore, the relationship between the partial correlation coefficient and spatial reduction is defined as follows: when the partial correlation coefficient is 0, the variable space is not reduced, and the space reduction is 0% of the initial space; when the partial correlation coefficient is 1, the upper (lower) space is abandoned, and the space reduction is 50% of the initial space.

## 3. Improved Particle Swarm Optimisation

### 3.1. Particle Swarm Optimisation

In the PSO algorithm, each particle contains three pieces of information: velocity, location, and fitness. Of these, the velocity information determines the direction change of a particle; the location information contains the applicable values of parameters; the fitness represents the performance of a particle. The updated equations of particle velocity and particle position are shown in Equations (6) and (7), respectively.

$$v_j^i = w \cdot v_j^{i-1} + c_1 r_1 (pbest_j - p_j^{i-1}) + c_2 r_2 (gbest - p_j^{i-1}) \tag{6}$$

$$p_j^i = p_j^{i-1} + v_j^i \tag{7}$$

where $v_j^i$ is the velocity of the *i*-th generation, *j*-th particle; $p_j^i$ is the position of the *i*-th generation, *j*-th particle; $w$ is the weighting factor, whose magnitude affects the inertia of the particle flight; $c_1$ and $c_2$ are learning factors, affecting the local and global fusion effect, respectively; $r_1$ and $r_2$ represent random numbers from 0 to 1; *Pbest* represents the optimal position information of the particles and records the optimal position of individual particles; and *gbest* represents the global optimal position information and records the

optimal position of the particle group. In our PSO algorithm, the value of $w$ was 0.8; $c_1 = c_2 = 2$.

### 3.2. Improvement of Particle Swarm Optimisation

In this study, the PSO algorithm is improved with respect to data processing of particle swarm population initialisation, data processing of iterative optimisation, particle velocity adjustment, and particle cross-boundary configuration.

#### 3.2.1. Data Processing of Particle Swarm Population Initialisation

The PSO algorithm generally uses Equation (8) for particle initialisation [26,39,40].

$$\begin{cases} p_j^0 = rand \times (p_{\max} - p_{\min}) + p_{\min} \\ v_j^0 = rand \times (v_{\max} - v_{\min}) + v_{\min} \end{cases} \tag{8}$$

where *rand* is a random number uniformly distributed between 0 and 1; $p_{max}$ and $p_{min}$ are the upper and lower boundary values of the particle position component, respectively; and $v_{max}$ and $v_{min}$ are the upper and lower boundary values of the particle velocity component, respectively.

Figure 4 illustrates the examples of random sampling applied to the 6-dimensional and 10-dimensional optimization problems, wherein two variables are selected for two-dimensional (2D) projection. We observed that data clustering or data omission existed in the sample space. Furthermore, uniform design (UD) was used to improve the initialization of the PSO algorithm to ensure uniformity of the initial particles in the space.

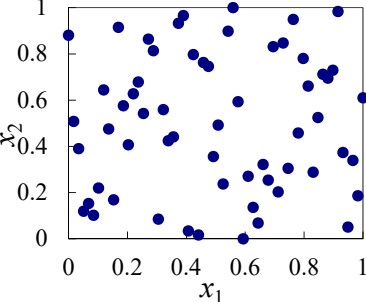

(**a**) 6-dimensional

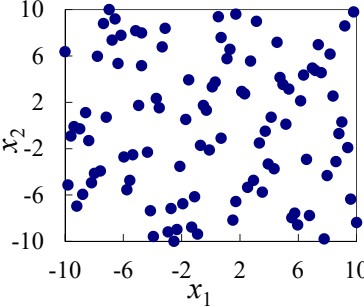

(**b**) 10-dimensional

**Figure 4.** Examples of random sampling.

UD, proposed by Fang [41] in China, is an experimental design method developed using the uniform distribution theory. In the uniform test, the level of each factor was evenly distributed within the test range, and each level was tested only once. Figure 5 depicts the projection of the sample points generated by the UD on a two-dimensional

plane. We observed that the distribution of sample points was uniform regardless of the dimensionality of the design variables.

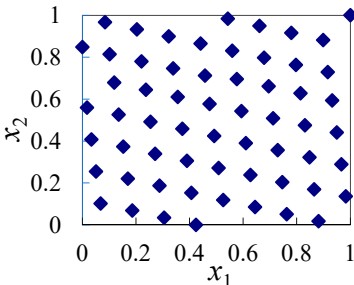

(**a**) 6-dimensional

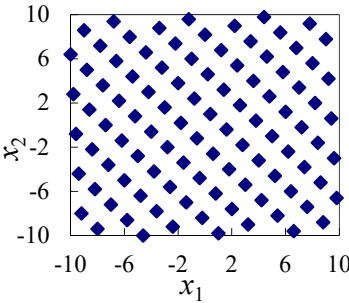

(**b**) 10-dimensional

**Figure 5.** Examples of uniform design sampling.

3.2.2. Data Processing of Particle Iterative Optimisation

The optimisation data in the iterative optimisation process also contain many hidden relationships that cannot be visually displayed. The optimisation process could generate a large amount of data. The analysis of these data can provide guidance to the subsequent particle optimisation, making the particles quickly cluster into a space worth exploring. So, this study used partial correlation analysis to perform datamining on the particles and the fitness values and thus obtain their relationships and complete the reduction of the initial search space of the particles. The partial correlation coefficient was calculated by Equations (1)–(3). According to the conclusions drawn in Section 2.2, when the partial correlation coefficient is set to 0, the range of the relevant design variables is not reduced; when the partial correlation coefficient is 1, the values of the relevant design variables are 50% of those in the initial range. In this article, two different methods are proposed to establish the relationship between the partial correlation coefficient and the degree of space reduction, which are the segmentation function reduction method as shown in Table 2 and the linear function reduction method as shown in Equation (9), where, $R_{ij}$ represents the partial correlation coefficient, and $Coe_{rp}$ represents the degree-of-space reduction.

$$Coe_{rp} = 50\% \times R_{ij} \qquad (9)$$

**Table 2.** Partial correlation coefficient and space reduction.

| $R_{ij}$ (Absolute Value) | 0.00–0.05 | 0.05–0.15 | 0.15–0.25 | 0.25–0.35 | 0.35–0.45 | 0.45–0.55 | 0.55–0.65 | 0.65–0.75 | 0.75–0.85 | 0.85–0.95 | 0.95–1.00 |
|---|---|---|---|---|---|---|---|---|---|---|---|
| $Coe_{rp}$ | 0% | 5% | 10% | 15% | 20% | 25% | 30% | 35% | 40% | 45% | 50% |

### 3.2.3. Particle Velocity Adjustment Strategy

As the particle search space changes, the speed of the particle swarm must be adjusted accordingly.

In the PSO algorithm, the upper and lower boundaries of the velocity are calculated by Equation (10), and the boundary values remain unchanged during the entire optimisation process. In this study, the value of $n$ is 5.

$$\begin{cases} v_{\max} = (p_{\max} - p_{\min})/n \\ v_{\min} = -(p_{\max} - p_{\min})/n \end{cases} \quad n \text{ is a positive integer} \tag{10}$$

The upper and lower boundaries of the velocity determine the accuracy of the area between the current position and the optimal position. If the velocity is too high, it may cause the particles to cross the optimal space. If it is too low, it may cause the particles to fall into a local optimum. Therefore, as the search space continues to decrease, the upper and lower boundaries of the particle velocity should change correspondingly with the search space. The upper and lower boundary values of the particle velocity in the optimisation process are modified as Equation (11).

$$\begin{cases} v_{\max}^i = (p_{\max}^i - p_{\min}^i)/n \\ v_{\min}^i = -(p_{\max}^i - p_{\min}^i)/n \end{cases} \quad n \text{ is a positive integer} \tag{11}$$

where $v_{\max}^i$ and $v_{\min}^i$ represent the optimisation velocity of the upper and lower boundaries in the $i$-th generation, and $p_{\max}^i$ and $p_{\min}^i$ represent the upper and lower boundary values of the particle position components in the $i$-th generation, respectively.

### 3.2.4. Particle Cross-Boundary Configuration

Because the search space of the particles is reduced during the optimisation process, the position and velocity of the particles often cross the upper and lower boundaries. In the PSO algorithm, these particles are set as the boundary values to resolve this problem. To give the particles better ability to explore the optimal solution in the reduced search space, our study configured the cross-boundary particles as shown in Equation (12).

$$\begin{cases} p_{\text{over}}^i = rand \times (p_{\max}^i - p_{\min}^i) + p_{\min}^i \\ v_{\text{over}}^i = rand \times (v_{\max}^i - v_{\min}^i) + v_{\min}^i \end{cases} \tag{12}$$

where $p_{over}^i$ and $v_{over}^i$ represent the position and velocity of the $i$-th generation cross-boundary particle; $v_{\max}^i$ and $v_{\min}^i$ represent the optimisation speed of the upper and lower boundaries of the $i$-th generation; $p_{\max}^i$ and $p_{\min}^i$ represent the upper and lower boundary values of the particle position components in the $i$-th generation, respectively; and $rand$ is a random number between 0 and 1.

### 3.3. Optimisation Framework of Improved Particle Swarm Optimisation Algorithm

The flowchart of the improved PSO algorithm is shown in Figure 6. The dashed box indicates improvement measures. The specific steps are as follows:

(1) input the parameters such as the number of particles and the number of iterations at the start of the algorithm;
(2) perform the initialisation process by randomised sampling using the particle swarm algorithm;
(3) perform data mining on the initialisation data and perform space reduction while completing the particle velocity adjustment;
(4) configure the cross-boundary particles in the iterative optimisation process, and perform data mining and space reduction after reaching a certain number of iterations;
(5) determine whether the optimisation is terminated by the maximum number of iterations.

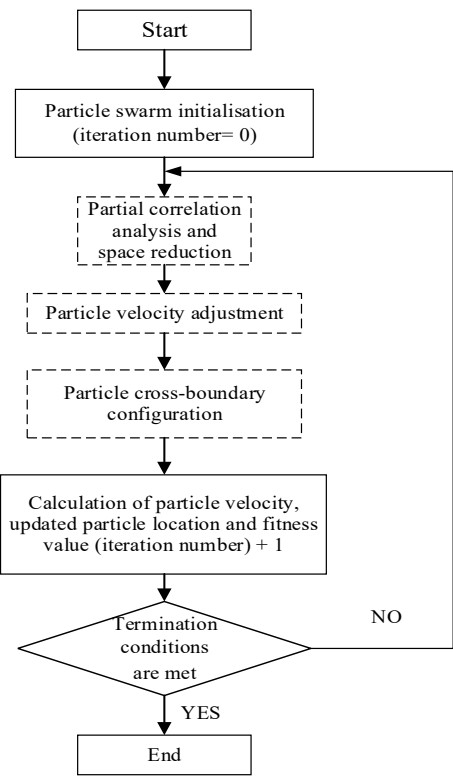

**Figure 6.** Flowchart of improved particle swarm optimisation algorithm.

The above improvement measures could improve the global and local optimisation ability of the PSO algorithm, which stays away from the local optimal solution and can quickly locate the optimisation space worthy of attention and improve the optimisation performance.

## 4. Function Examples of the Improved Particle Swarm Optimisation Algorithm

Four commonly used numerical functions were selected to test the performance of the algorithm. To facilitate the drafting of the optimised iteration convergence graph, the numerical results of the four selected numerical functions were all added by 1 based on the corresponding reference numerical functions. The expressions of the four numerical functions are shown in Equations (13)–(16), and the function name, test dimension, variable range, theoretical optimal scheme, and corresponding optimal solution are also given.

(1) *Levy* function ($n = 5, -10 \leq x_i \leq 10, i = 1, \cdots, n$)

$$H_1(x) = 10\ \sin^2(\pi x_1) + \sum_{i=1}^{n-1} 100\ xi(1 + 10\ \sin^2(\pi x_{i+1})) + 100\ (x_n - 2)^2 + 1 \tag{13}$$

where $x^* = (0, \cdots, 0, 1)^{\mathrm{T}}$, $H_1(x^*) = 1$.

(2) *Trigonometric* function ($n = 5, -10 \leq x_i \leq 10, i = 1, \cdots, n$)

$$H_2(x) = \sum_{i=1}^{n} [8\sin^2(7(x_i - 0.9)^2) + 6\sin^2(14(x_i - 0.9)^2) + (x_i - 0.9)^2] + 1 \tag{14}$$

where $x^* = (0.9, \cdots, 0.9)^{\mathrm{T}}$, $H_2(x^*) = 1$.

(3) *Griewank* function ($n = 5, -10 \leq x_i \leq 10, i = 1, \cdots, n$)

$$H_3(x) = \frac{1}{4000}\sum_{i=1}^{n} x_i^2 - \prod_{i=1}^{n} \cos(\frac{x_i}{\sqrt{i}}) + 2 \tag{15}$$

where $x^* = (0, \cdots, 0)^T$, $H_3(x^*) = 1$.
(4) *Pinter* function $(n = 5, -10 \le x_i \le 10, i = 1, \cdots, n)$

$$H_4(x) = \sum_{i=1}^{n} i x_i^2 + \sum_{i=1}^{n} 20\, i \sin^2(x_{i-1} \sin x_i - x_i + \sin x_{i+1}) + \sum_{i=1}^{n} i \log_{10}(1 + i(x_{i-1}^2 - 2x_i + 3x_{i+1} - \cos x_i + 1)^2) + 1 \quad (16)$$

where $x_0 = x_n$, $x_{n+1} = x_1$, $x^* = (0, \cdots, 0)^T$, $H_4(x^*) = 1$.

Figure 7 is a schematic diagram showing the spatial structure of the two-dimensional performance of each function.

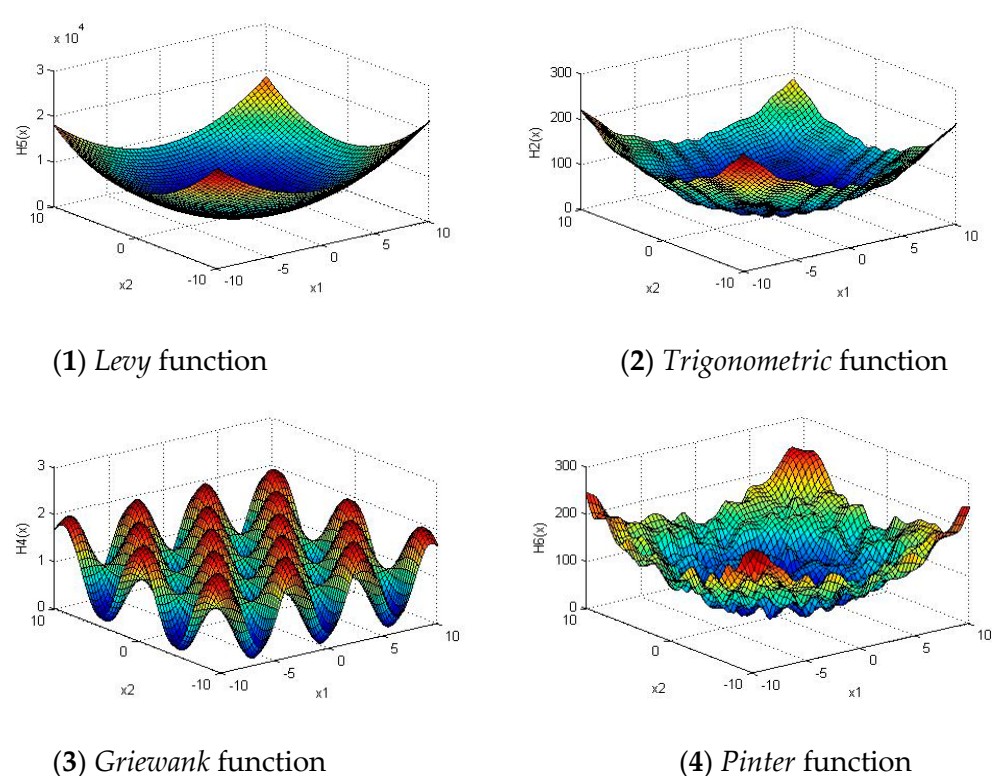

(**1**) *Levy* function                    (**2**) *Trigonometric* function

(**3**) *Griewank* function                    (**4**) *Pinter* function

**Figure 7.** Schematic diagram showing the spatial structure of the two-dimensional performance of each function.

In the function examples, both the PSO algorithm and improved PSO algorithm were applied for optimisation, and the optimised particle number was set to 100. The number of iterations was 50 generations. The space reduction methods used in the improved PSO algorithm were the segmentation reduction method and the linear reduction method mentioned in Section 3.2.2, respectively. Because the initial range of function variables was large, space reduction was set to be conducted every other generation. The optimisation iteration convergence graphs of the four functions are shown in Figure 6, and the optimisation results are shown in Table 3. Convergent algebra denotes the rate of convergence. In this paper, the algebra is used as a convergence standard. In Figure 8, the *x*-axis is the number of iterations, and the *y*-axis is the optimal value of the current iteration. The different line types represent different optimisation algorithms (where improved PSO algorithm 1 is the segmentation reduction method, and improved PSO algorithm 2 is the linear reduction method).

**Table 3.** Optimised results of function algorithms.

| Function Name | Optimisation Method | Convergent Algebra | Optimisation Optimal Value | Theoretical Optimal Value |
|---|---|---|---|---|
| *Levy* | PSO | 12 | 1.296 | |
| | Improved PSO 1 | 11 | 1.250 | 1 |
| | Improved PSO 2 | 11 | 1.001 | |
| *Trigonometric* | PSO | 39 | 2.332 | |
| | Improved PSO 1 | 21 | 1.000 | 1 |
| | Improved PSO 2 | 20 | 1.000 | |
| *Griewank* | PSO | 28 | 1.026 | |
| | Improved PSO 1 | 16 | 1.000 | 1 |
| | Improved PSO 2 | 10 | 1.000 | |
| *Pinter* | PSO | 35 | 1.423 | |
| | Improved PSO 1 | 18 | 1.050 | 1 |
| | Improved PSO 2 | 19 | 1.011 | |

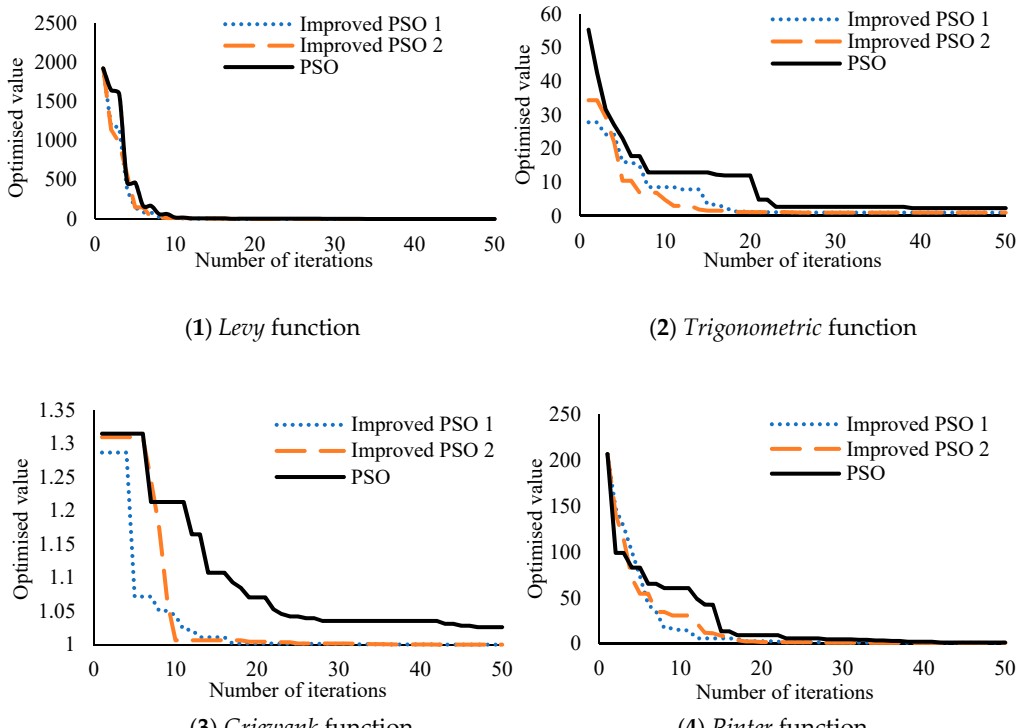

**Figure 8.** Optimised iterative convergence plot.

As seen in Figure 8 and Table 3, for the Levy function with a simple space, both the improved PSO algorithm and the PSO algorithm can find the optimal solution in a short time; thus, the space reduction method did not significantly improve the efficiency. For the Griewank function with a complex space, there are many local optimal solutions in the function, making the PSO algorithm fall into local optima, whereas the improved PSO algorithm can obtain the optimal solution. For the trigonometric function and the Pinter function, the improved PSO algorithm can improve the overall efficiency and performance of the optimisation. According to the specific optimisation results listed in Table 3, all optimal values obtained by the improved PSO algorithm were better than those of the PSO algorithm and were closer to the theoretical optimal value.

To demonstrate the robustness of algorithm, each function was optimized using both the improved PSO algorithm and PSO algorithm 100 times.

The obtained global optimum for each run is illustrated in Figure 9 using boxplots. A smaller or shorter box implies that the standard deviation was small, and the symbol (red points) denotes the abnormal value. The relatively smaller boxes of DSROF for most test functions validate that its robustness was higher than that of PSO, particularly in the case of complex functions. For the Levy function, both the improved PSO algorithm and the PSO algorithm can find the optimal solution. However, the standard deviation and average of the improved PSO are only a little bigger than PSO.

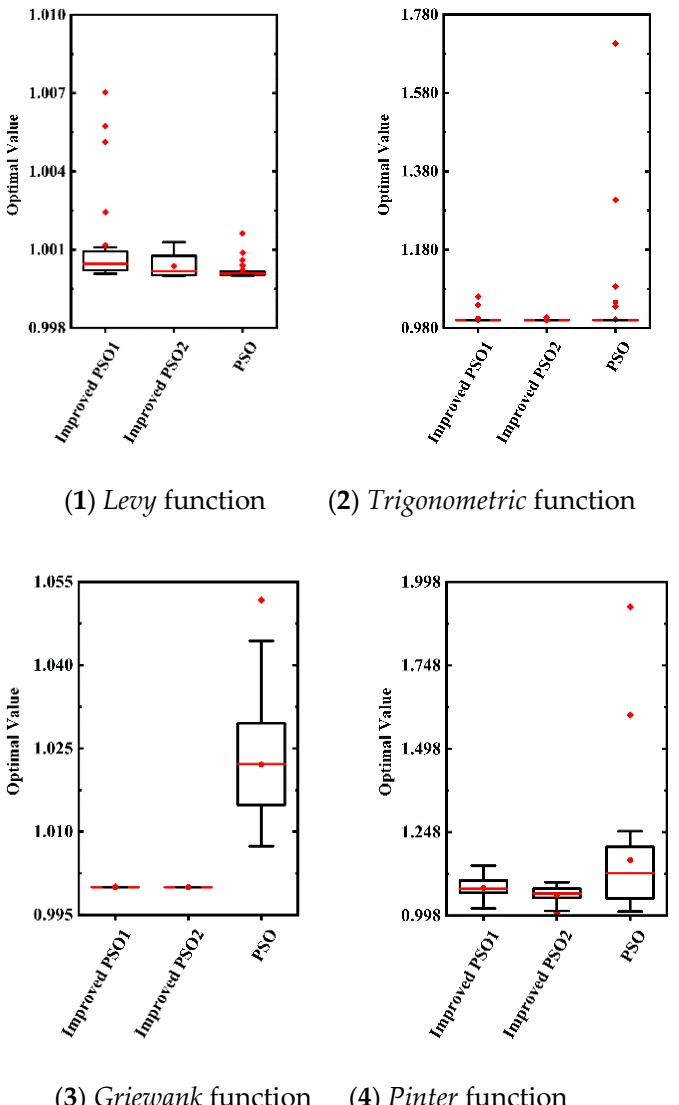

(**1**) *Levy* function     (**2**) *Trigonometric* function

(**3**) *Griewank* function     (**4**) *Pinter* function

**Figure 9.** Boxplot of the best solution for all test problems.

## 5. Bow Shape Optimisation in Engineering Vessel

According to the research described in Section 3, the linear function was used to establish the relationship between the partial correlation coefficient and the degree of space reduction. In this study, an engineering vessel was used to optimise the bow shape to reduce the wave-making resistance. A scale ratio of 1:20 was used to model the hull form. The main parameters of the model are shown in Table 4. The front of the model has an invisible bulbous bow. The model is shown in Figure 10.

**Table 4.** Major parameters of an engineering vessel.

| Length between Perpendiculars $L_{pp}$ (m) | Waterline Width $B_{wl}$ (m) | Draft $T$ (m) | Block Coefficient $C_b$ | Drainage Volume $\nabla$ (m³) | Wet Surface Area $S_{wet}$ (m²) | Floating Centre Longitudinal Position $L_{cb}$ (m) |
|---|---|---|---|---|---|---|
| 4.995 | 0.770 | 0.255 | 0.674 | 0.646 | 4.764 | 2.518 m |

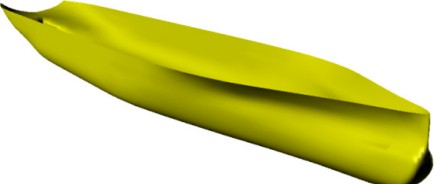

**Figure 10.** Hull model.

*5.1. Definition of Optimisation*

Optimisation Problem Definition

Three changeable control points were selected as the optimisation variables at the bow waterline position and the bilge position of the hull, which were numbered $X_1$–$X_6$. The distribution of these data in the hull is shown in Figure 11. The change direction of all optimisation variables was in the ship width (y) direction. The range of variables and initial values are shown in Table 5.

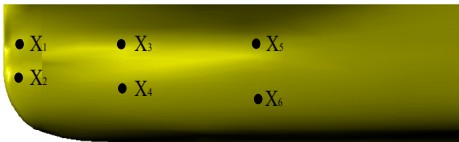

**Figure 11.** Distribution of the optimisation variables.

**Table 5.** Range of the optimisation variables.

| Optimisation Variable | Lower Limit | Upper Limit |
|---|---|---|
| $Y_1$ | 0.01000 | 0.01300 |
| $Y_2$ | 0.04000 | 0.06000 |
| $Y_3$ | 0.07200 | 0.09700 |
| $Y_4$ | 0.08500 | 0.12500 |
| $Y_5$ | 0.20000 | 0.24000 |
| $Y_6$ | 0.17000 | 0.22000 |

The optimisation target is the wave-making resistance coefficient **min** $C_w$ of the ship. The Froude number during ship optimisation was $F_n = 0.24$. The optimisation considered only the conditions of full restrictions, regardless of heave and pitch motions. To ensure that the performance of the ship does not change too much under certain drainage conditions, the drainage volume ($\Delta$), and the longitudinal position of the floating centre ($L_{cb}$) were selected as the hydrostatic constraints. The specific settings are as follows.

$$0\% \leq \frac{\Delta_{\text{opt}} - \Delta_{\text{initial}}}{\Delta_{\text{initial}}} \leq 1\%$$

$$\left| \frac{L_{cb\text{opt}} - L_{cb\text{initial}}}{L_{cb\text{initial}}} \right| \leq 1\%$$

Because only the bow form was optimised, and the layout was at the middle section of the ship, the tail form and the contour of the hull were constrained to keep the shape of the rear hull and the hull profile stable during the optimisation process. The hull constraint points are arranged as shown by the red line in Figure 12.

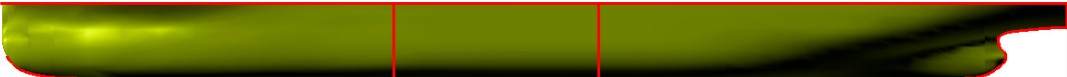

**Figure 12.** Arrangement of hull constraint points.

The improved PSO algorithm and the PSO algorithm were applied to the engineering vessel for optimisation. The population data generated by the initialisation were consistent. Because there are many constraints in hull form optimisation, several infeasible solutions can be generate. This makes it more complex than the function examples. Moreover, the optimisation space value of the hull variable is small. To ensure the accuracy of datamining, the partial correlation analysis must be performed every four generations, along with space reduction.

### 5.2. Flowchart of Hull Form Optimisation

Figure 13 is a flowchart of hull form optimisation. In our study, we used the hull surface modification method based on radial basis function interpolation [9,42,43] to obtain the new hull form. The commercial software Shipflow was used to calculate the ship wave-making resistance performance. The entire optimisation process and calculations were performed on a self-developed multi-disciplinary integrated optimisation platform for ship hydrodynamic performance [44]. The number of particles was set to 60, and the optimisation iteration was set to 20 generations. The computer used for the calculation was a 2.9-GHz Intel i9-7920x with 32 GB of RAM, running the Windows 7 operating system.

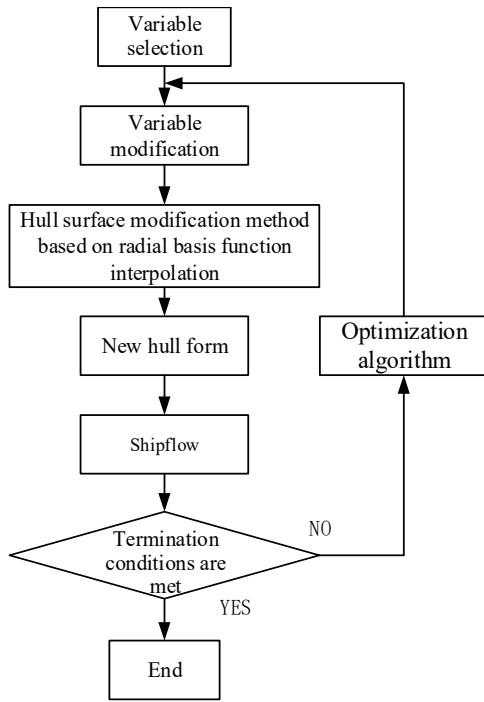

**Figure 13.** Flowchart of hull form optimisation with dynamic space reduction.

### 5.3. Hull Form Optimisation Results

The optimised iterative convergence graph is shown in Figure 14. The comparison shows that the improved PSO algorithm tended to converge after 11 generations, whereas the PSO algorithm tended to converge after the 17th generation. The time for the two algorithms to reach the maximum number of iterations was approximately 20 h. The time for the improved PSO algorithm to reach convergence was 11 h 30 min. The time for the PSO algorithm to reach convergence was 16 h. Therefore, the improved PSO algorithm can improve the optimisation efficiency in the example of hull form optimisation.

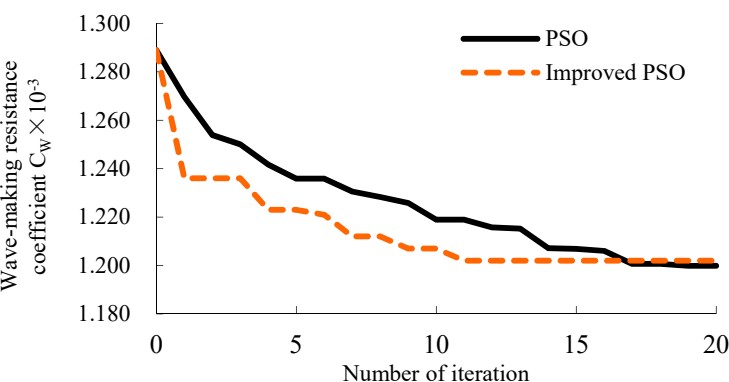

**Figure 14.** Optimised iterative convergence graph.

Table 6 lists the variable data comparison between the PSO algorithm (Opt1) and the improved PSO algorithm (Opt2). It shows that the optimised hull form obtained by both algorithms was basically the same. The optimised wave-making resistance coefficients were also similar. The wave-making resistance coefficient of Opt1 was reduced by 10.24%, and the wave-making resistance coefficient of Opt2 was reduced by 10.10%. These results indicated that the improved PSO algorithm can ensure satisfactory performance in the optimisation process.

**Table 6.** Comparison of optimisation results.

| Parameter | $X_1$ | $X_2$ | $X_3$ | $X_4$ | $X_5$ | $X_6$ | $Cw \times 10^3$ | Change |
|---|---|---|---|---|---|---|---|---|
| Initial hull | 0.0116 | 0.0474 | 0.0859 | 0.1013 | 0.1938 | 0.2183 | 1.337 | 0% |
| Opt1 | 0.0107 | 0.0600 | 0.0720 | 0.0959 | 0.1941 | 0.2127 | 1.200 | −10.24% |
| Opt2 | 0.0112 | 0.0600 | 0.0722 | 0.0939 | 0.1939 | 0.2157 | 1.202 | −10.10% |

Table 7 lists the changes in the major parameters of the optimised hull and the initial hull using the two optimisation algorithms at *Fn* = 0.24. The comparison shows that the drainage volume and the longitudinal position of the floating centre of Opt1 and Opt2 were essentially the same as those of the initial hull. The wet surface area was slightly increased compared with the initial hull. The total drag coefficient of Opt1 decreased by 3.07%, and the total drag coefficient of Opt2 decreased by 2.96%. Considering the errors in CFD calculations, the hull form results obtained by the two optimisation algorithms were basically the same. The total drag coefficient was calculated by Shipflow.

**Table 7.** Comparison of optimisation results at *Fn* = 0.24.

| | Floating Centre Longitudinal Position (m) | Drainage Volume (m³) | Wet Surface Area (m²) | Total Drag Coefficient $C_t \times 10^3$ | Change |
|---|---|---|---|---|---|
| Initial hull | 2.518 | 0.646 | 4.764 | 4.909 | 0 |
| Opt1 | 2.520 | 0.646 | 4.774 | 4.758 | −3.07% |
| Opt2 | 2.519 | 0.646 | 4.773 | 4.764 | −2.96% |

The comparison between Tables 6 and 7 shows that the hull form obtained by both the standard and improved PSO algorithms was consistent. Opt1, Opt2, and the initial hull were subjected to profile analyses. Figures 15 and 16 provide a comparison of the cross-sectional line and the vertical line of the initial hull and the optimised hull. A comparison of the cross-sectional lines shows that the contour of the two optimised hulls close to the invisible bullnose was slightly increased compared with the initial hull, making the bow profile smooth, reducing the ridge vortex, and thus reducing the drag. The waterline from the bow to the middle section was slightly contracted compared with the initial hull.

The bow waterline was more pointed and thinner, which reduces the inflow angle and is beneficial to reduce the wave-making resistance of the hull.

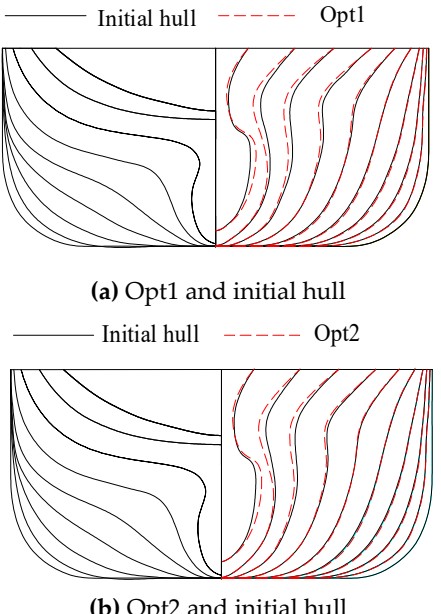

**(a)** Opt1 and initial hull

**(b)** Opt2 and initial hull

**Figure 15.** Cross-sectional line comparison chart.

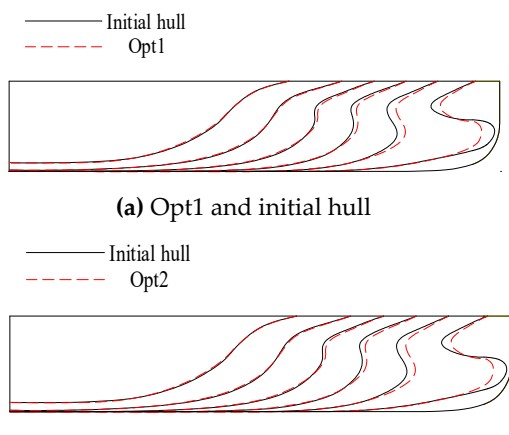

**(a)** Opt1 and initial hull

**(b)** Opt2 and initial hull

**Figure 16.** Vertical line comparison chart.

Figures 17 and 18 provide a comparison of the wave profile along the hull and waveform plots of Opt1, Opt2, and the initial hull. The amplitude of the optimised hull was lower than that of the prototype hull, and the amplitude around the hull was significantly lower. The wave system of the optimised hull has become simpler, leading to a decline in the wave-making resistance. Figure 19 shows the hull pressure distribution before and after optimisation. A comparison with the prototype shows that the optimised profile reduces the positive and negative pressures at the bow and tail of the hull, thereby reducing the drag.

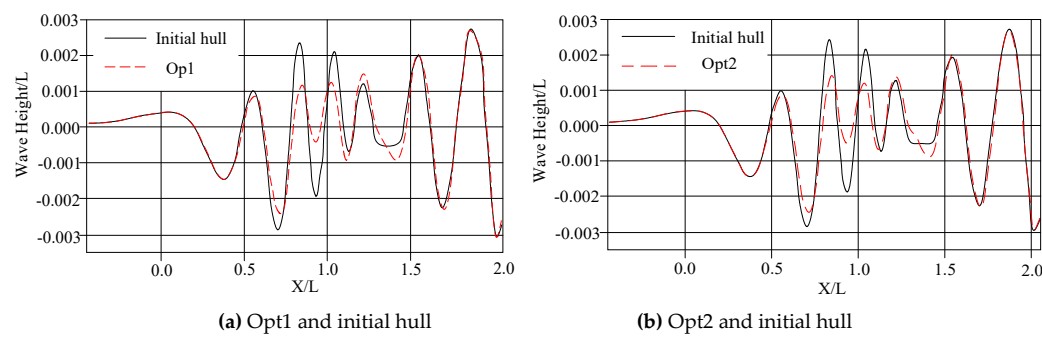

**(a)** Opt1 and initial hull

**(b)** Opt2 and initial hull

**Figure 17.** Wave profile along the hull at *Fn* = 0.24 (*y*/L = −0.301).

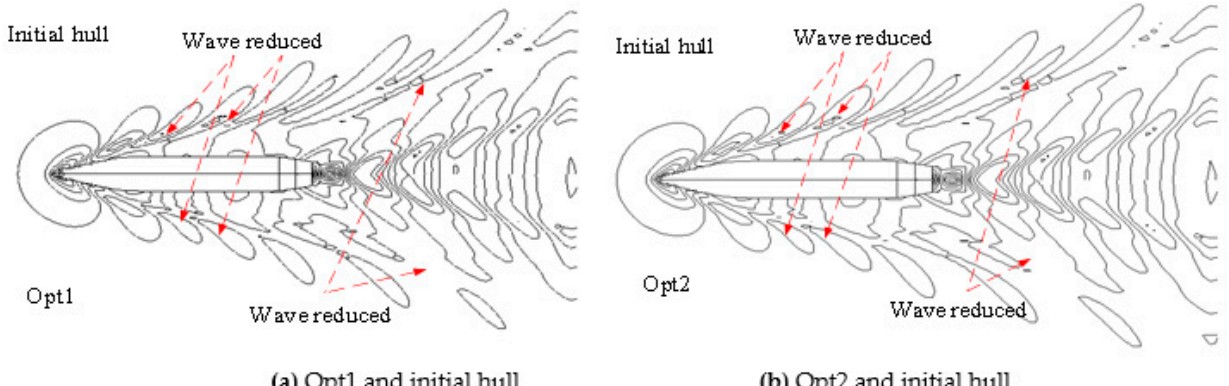

**(a)** Opt1 and initial hull

**(b)** Opt2 and initial hull

**Figure 18.** *Fn* = 0.24 waveform.

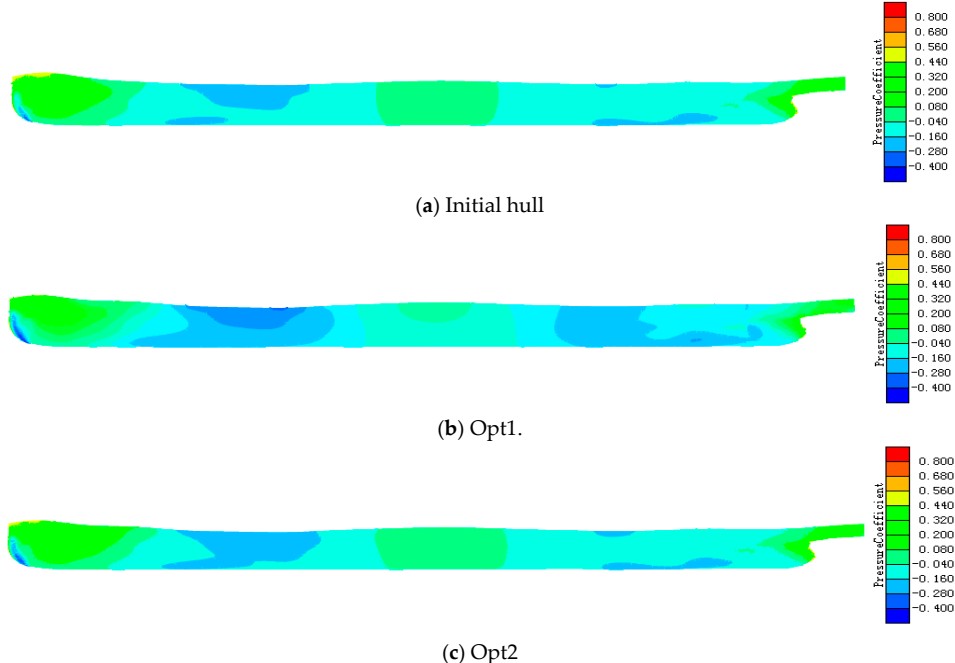

(**a**) Initial hull

(**b**) Opt1.

(**c**) Opt2

**Figure 19.** *Fn* = 0.24 hull pressure distribution map.

## 6. Conclusions

The PSO algorithm has the problems of local premature convergence and low convergence performance. Our study combined the space reduction techniques and improved the algorithm from different aspects, including data processing of particle swarm population initialisation, data processing of iterative optimisation, particle velocity adjustment, and particle cross-boundary configuration. Commonly used function examples were optimised

to verify the feasibility of the improved PSO algorithm. Finally, our algorithm was applied to the hull form optimisation of an engineering vessel, and the optimisation result was compared with that of the PSO algorithm. The following conclusions can be drawn from our study:

(1) For the optimisation of a simple space, the improved PSO algorithm did not significantly enhance the optimisation efficiency and performance compared with the PSO algorithm, and both algorithms can obtain fast convergence results. For more complex optimisation problems, PSO more easily falls into the local optimal solution, whereas the improved PSO algorithm can avoid the local optimum owing to datamining of the optimised data in the optimisation process, which can provide guidance on the optimisation of subsequent particles.

(2) The application of the algorithm in engineering practice is verified by hull form optimisation. This algorithm can improve the optimisation efficiency to a certain extent while ensuring high performance, thus reducing the overall time of hull form optimisation. This has certain value in engineering applications.

(3) Our study used partial correlation analysis for datamining. Because the coefficient obtained by a partial correlation analysis cannot directly perform space reduction, a certain relationship must be established. For optimisation with too many iterations, the segmentation reduction method and the linear reduction method may lose the optimal solution. In the particle initialisation stage, the particle information obtained cannot be evenly distributed in the optimisation space. As a result, the optimisation information obtained by the previous datamining is not accurate, leading to reduced optimisation efficiency. Further research is required to address these issues.

**Author Contributions:** Conceptualization: Q.Z., Z.-Y.L.; Data Curation: Q.Z.; Formal analysis: Q.Z., Z.-Y.L.; Validation: Q.Z., H.-C.C.; Methodology: Q.Z., B.-W.F. All authors have read and agreed to the published version of the manuscript.

**Funding:** This research was funded by the National Natural Science Foundation of China [grant numbers 551720105011, 51979211], Research on the Intelligentized Design Technology for Hull Form. Green Intelligent Inland Ship Innovation Programme. Research on the Design of Large-scale Marine Tourism Floating Complex. The Fundamental Research Funds for the Central Universities (2020-YB-016). High-tech ship research project (2019[357]).

**Institutional Review Board Statement:** Not applicable.

**Informed Consent Statement:** Not applicable.

**Data Availability Statement:** The data presented in this study are available in this article (Tables and Figures).

**Conflicts of Interest:** The authors declare no conflict of interest.

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
