# Peer review of "Application of Improved Particle Swarm Optimisation Algorithm in Hull form Optimisation"

_jmse, doi:10.3390/jmse9090955_

Round 1

Reviewer 1 Report

This is very interesting paper dealing with the important topic of the hull form optimization. The paper is devided into two parts. First part deals with the improvement of the optimization method, while the second part is the application of the optimization on the hull form optimization.The paper is nicely written and easy to follow. My opinion iis that the paper is of sufficient quality to be accepted for publication. However, some improvements in the manuscript should be done as precised int he comments below.   

The present reviewer doesn't feel qualified to judge about details of the optimization method, although it looks quite convincing that authors succeded  to improve the method. The authors presented enough results and evidences showing improved performances of the PSO method. One comments is that the caption of Figure 6 is missing. 

Regarding second part of the paper,  i.e. hull optimization, it would be necessary that the authors explain clearly in the text why they have used the ship model instead of the full-size ship. 

Authors should explain in the text meaning of the non-dimensional  optimization variables  Y1-Y6 that is missing. 

Author Response

Reviewer #1

Point1: This is very interesting paper dealing with the important topic of the hull form optimization. The paper is devided into two parts. First part deals with the improvement of the optimization method, while the second part is the application of the optimization on the hull form optimization.The paper is nicely written and easy to follow. My opinion iis that the paper is of sufficient quality to be accepted for publication. However, some improvements in the manuscript should be done as precised int he comments below.

Reply1: Thanks for your careful work and thoughtful suggestions that have helped improve this paper substantially.

Point2: The present reviewer doesn't feel qualified to judge about details of the optimization method, although it looks quite convincing that authors succeded to improve the method. The authors presented enough results and evidences showing improved performances of the PSO method. One comments is that the caption of Figure 6 is missing.

Reply2: It has been added in the article and has been marked.

Point3: Regarding second part of the paper, i.e. hull optimization, it would be necessary that the authors explain clearly in the text why they have used the ship model instead of the full-size ship.

Reply3: In order to reduce the number of grids and computational speeds of CFD simulation, the ship model is used in the simulation process. The Froude number of ship model is equal to full-size ship to ensure that the flow field is similar.

Point4: Authors should explain in the text meaning of the non-dimensional optimization variables Y1-Y6 that is missing.

Reply4: Six changeable control points are selected in ship bow and numbered as X1-6. Where, X1, X3, X5 control the shape of bow waterline. X2, X4, X6 control the shape of bow bilge.

Reviewer 2 Report

The authors present an improved PSO for shape optimization in naval engineering design problems. My main concern is about the awareness of the authors about many recent works in the field that should be included in the literature review to provide to the reader what are the benefits of the proposed method compared to other methods. In the following some relevant works that have to be considered and included in the paper discussion. In particular, some of them are focused on the use of a deterministic variant of PSO that is quite effective in this context. The authors should provides comparison with this method.

- D'Agostino D, Serani A, Diez M (2020) Design-space assessment and dimensionality reduction: an of-line method for shape reparameterization in simulation-based optimization. Ocean Eng 197:106852
- Diez M, Campana EF, Stern F (2018) Stochastic optimization methods for ship resistance and operational eiciency via CFD. Struct Multidiscip Optim 57(2):735-758
- Grigoropoulos G, Campana E, Diez M, Serani A, Goren O, Sariöz K, DaniÅŸman D, Visonneau M, Queutey P, Abdel-Maksoud M, et al. (2017) Mission-based hull-form and propeller optimization of a transom stern destroyer for best performance in the sea environment. In: VII International conference on computational methods in marine engineering MARINE2017
- Harries S, Abt C (2019) Faster turn-around times for the design and optimization of functional surfaces. Ocean Eng 193:106470
- He, P., Filip, G., Martins, J. R., & Maki, K. J. (2019). Design optimization for self-propulsion of a bulk carrier hull using a discrete adjoint method. Computers & Fluids, 192, 104259.
- Lin Y, He J, Li K (2018) Hull form design optimization of twinskeg ishing vessel for minimum resistance based on surrogate model. Adv Eng Softw 123:38-50
- Miao A, Zhao M, Wan D (2020) CFD-based multi-objective optimisation of S60 catamaran considering demihull shape and separation. Appl Ocean Res 97:102071
- Serani, A., Stern, F., Campana, E. F., & Diez, M. (2021). Hull-form stochastic optimization via computational-cost reduction methods. Engineering with Computers, 1-25.

- Serani, A., Leotardi, C., Iemma, U., Campana, E. F., Fasano, G., & Diez, M. (2016). Parameter selection in synchronous and asynchronous deterministic particle swarm optimization for ship hydrodynamics problems. Applied Soft Computing, 49, 313-334.

- Serani, A., Fasano, G., Liuzzi, G., Lucidi, S., Iemma, U., Campana, E. F., ... & Diez, M. (2016). Ship hydrodynamic optimization by local hybridization of deterministic derivative-free global algorithms. Applied Ocean Research, 59, 115-128.

- Tezdogan T, Shenglong Z, Demirel YK, Liu W, Leping X, Yuyang L, Kurt RE, Djatmiko EB, Incecik A (2018) An investigation into ishing boat optimisation using a hybrid algorithm. Ocean Eng 167:204-220

- Tezzele M, Salmoiraghi F, Mola A, Rozza G (2018) Dimension reduction in heterogeneous parametric spaces with application to naval engineering shape design problems. Adv Model Simul Eng Sci 5(1):25

- Yang C, Huang F (2016) An overview of simulation-based hydrodynamic design of ship hull forms. J Hydrodyn Ser B 28(6):947-960

- Zhang S, Tezdogan T, Zhang B, Xu L, Lai Y (2018) Hull form optimisation in waves based on CFD technique. Ships Ofshore Struct 13(2):149-164

- Zhang S, Zhang B, Tezdogan T, Xu L, Lai Y (2018) Computational fluid dynamics-based hull form optimization using approximation method. Eng Appl Comput Fluid Mech 12(1):74-88

Other major issues:

1 - the method used is stochastic and needs statistical significant results to be presented. The authors should repeat at least 100 times each optimization and provides statistics, such as mean, standard deviation, and confidence interval

2 - the use of only 4 test functions with 2 dimensions is not sufficient, the authors should enrich the basket of benchmark problems, see in the reference list provided there are several examples, up to 50 dimensions.

3 - The description of the methodology is not clear and have to be improved

Author Response

Reviewer #2

Point1: The authors present an improved PSO for shape optimization in naval engineering design problems. My main concern is about the awareness of the authors about many recent works in the field that should be included in the literature review to provide to the reader what are the benefits of the proposed method compared to other methods. In the following some relevant works that have to be considered and included in the paper discussion. In particular, some of them are focused on the use of a deterministic variant of PSO that is quite effective in this context. The authors should provides comparison with this method.

Reply1: Thanks for your careful work and thoughtful suggestions that have helped improve this paper substantially. The relevant papers and scholars you mentioned, we have recently read and added reference in our paper. PSO is a stochastic optimization method. In ship design optimization problems, the deterministic version of PSO has been proven to be very effective. It contains an initial uniform distribution based on the Hammersley sequence sampling. We have read the relevant literature and are very interested in the deterministic version of PSO (DPSO) algorithm. In the future, we will conduct in-depth research on the DPSO.

Other major issues:

Point2: the method used is stochastic and needs statistical significant results to be presented. The authors should repeat at least 100 times each optimization and provides statistics, such as mean, standard deviation, and confidence interval.

Reply2: Each function was optimised another 100 times. Boxplot of the best solution for all test function has added in the article. Since only one week of modification, the CFD example in the article has not been optimized many times. So we still keep only 1 result for the CFD problem in the article. This is also convenient for better comparison of CFD results and moulded lines.

Point3: the use of only 4 test functions with 2 dimensions is not sufficient, the authors should enrich the basket of benchmark problems, see in the reference list provided there are several examples, up to 50 dimensions.

Reply3: In the article, four numerical functions are 5 dimensions. So we only selected 6-dimensional optimization in CFD example. If the modification time is enough, we will supplement the example.

In the follow-up work, we developed the Improved PSO2 and investigated a dynamic space reduction optimization framework (DSROF). We compared the method with PSO and NSGA2. Each function was optimised 20 times. The collected range and average values of the obtained best solution fopt are given in table 1.

Table 1 test result of fopt obtained by PSO, NSGA2 and DSROF

Fun.

PSO

NSGA2

DSROF

Collected range

fopt

Collected range

fopt

Collected range

fopt

SFNO.2

[0.0001,0.0049]

0.0013

[0.0025,0.0432]

0.0323

[0.0001,0.0001]

0.0001

SC

[-1.0316,-1.0312]

-1.0315

[-1.0316,-1.0312]

-1.0314

[-1.0316,-1.0030]

-1.0304

MF

[-1.9132,-1.9131]

-1.9132

[-1.9131,-1.9130]

-1.9130

[-1.9132,-1.9132]

-1.9132

GF

[0.5233,1.0884]

0.6352

[0.5233,0.5236]

0.5235

[0.5233,0.5233]

0.5233

DW

[-0.9985,-0.9362]

-0.9659

[-1.0000,-0.9963]

-0.9977

[-1.0000,-0.9898]

-0.9959

SP3

[0.0002,0.0050]

0.0011

[0.0002,0.0004]

0.0002

[0.0000,0.0004]

0.0000

AF4

[0.0069,0.0436]

0.0219

[0.0072,0.0532]

0.0132

[0.0000,0.0057]

0.0012

HF6

[-3.3209,-3.1921]

-3.2704

[-3.3211,-3.3211]

-3.3211

[-3.2634,-3.3219]

-3.3067

TR6

[-49.8360,-47.0466]

-48.5274

[-49.6214,-48.2239]

-48.9226

[-48.9561,-46.3729]

-47.5442

TR10

[-209.957,-200.714]

-206.159

[-208.322,-204.596]

-207.600

[-208.921,-206.164]

-207.984

GN10

[0.0074,0.0935]

0.0423

[0.0689,0.0764]

0.0727

[0.0000,0.0000]

0.0000

PF10

[0.0522,0.4194]

0.2008

[0.0622,38.8894]

11.3421

[0.0389,0.0705]

0.0563

WS12

[0.0000,0.0000]

0.0000

[0.0000,0.0000]

0.0000

[0.0000,0.0000]

0.0000

RF16

[0.7671,1.2721]

0.8270

[1.0722,1.4701]

1.1537

[0.0486,0.4363]

0.1416

TF20

[0.4656,1.4937]

0.9533

[0.0832,0.9522]

0.2133

[0.1462,0.3822]

0.3949

Point4: The description of the methodology is not clear and have to be improved

Reply4: The methodology has been improved. And marked red in the article.

Reviewer 3 Report

The English form requires moderate editing, although the terminology is often inappropriate.

The reference list is often outdated, furthermore, in the context of PSO development the reviewer suggests the following references:

  • Serani, A., Leotardi, C., Iemma, U., Campana, E. F., Fasano, G., & Diez, M. (2016). Parameter selection in synchronous and asynchronous deterministic particle swarm optimization for ship hydrodynamics problems. Applied Soft Computing49, 313-334.
  • Leotardi, C., Serani, A., Diez, M., Campana, E. F., Fasano, G., & Gusso, R. (2021). Dense conjugate initialization for deterministic PSO in applications: ORTHOinit+. Applied Soft Computing104, 107121.

The objective and approach of the paper is not clearly stated. Specifically, it is not clear which of the PSO weaknesses the authors intend to address.

Since PSO makes use of random coefficients, a statistical validation of the results should be provided.

The proposed PSO version should be compared also with some PSO versions from literature and not only with the standard (and old) original PSO version.

The motivation for the selection of a specific set of PSO parameters should be motivated.

The authors state that “Considering the errors in CFD calculations, the hull form results obtained by the two optimisation algorithms were basically the same”, the question is: why anyone should use the proposed version is not any significant improvement is demonstrated?

Author Response

Reviewer #3

Point1: The English form requires moderate editing, although the terminology is often inappropriate.

Reply1: Thanks for your careful work and thoughtful suggestions that have helped improve this paper substantially.

Point2: The reference list is often outdated, furthermore, in the context of PSO development the reviewer suggests the following references:

  • Serani, A., Leotardi, C., Iemma, U., Campana, E. F., Fasano, G., & Diez, M. (2016). Parameter selection in synchronous and asynchronous deterministic particle swarm optimization for ship hydrodynamics problems. Applied Soft Computing49, 313-334.
  • Leotardi, C., Serani, A., Diez, M., Campana, E. F., Fasano, G., & Gusso, R. (2021). Dense conjugate initialization for deterministic PSO in applications: ORTHOinit+. Applied Soft Computing104, 107121.

Reply2: Thanks for your careful work and thoughtful suggestions that have helped improve this paper substantially. The relevant papers and scholars you mentioned, we have recently read and added reference in our paper.

Point3: The objective and approach of the paper is not clearly stated. Specifically, it is not clear which of the PSO weaknesses the authors intend to address.

Reply3: Similar to other intelligent algorithms, PSO also has the disadvantages of local premature convergence and low convergence performance. And optimization data are not used to analyse and reduce the range of values for relevant design variables

Point4: Since PSO makes use of random coefficients, a statistical validation of the results should be provided.

Reply4: Each function was optimised another 100 times. Boxplot of the best solution for all test function has added in the article.

Point5: The proposed PSO version should be compared also with some PSO versions from literature and not only with the standard (and old) original PSO version.

Reply5: In the follow-up work, we developed the Improved PSO2 and investigated a dynamic space reduction optimization framework (DSROF). We compared the method with PSO and NSGA2. Each function was optimised 20 times. The collected range and average values of the obtained best solution fopt are given in table 1.

Table 1 test result of fopt obtained by PSO, NSGA2 and DSROF

Fun.

PSO

NSGA2

DSROF

Collected range

fopt

Collected range

fopt

Collected range

fopt

SFNO.2

[0.0001,0.0049]

0.0013

[0.0025,0.0432]

0.0323

[0.0001,0.0001]

0.0001

SC

[-1.0316,-1.0312]

-1.0315

[-1.0316,-1.0312]

-1.0314

[-1.0316,-1.0030]

-1.0304

MF

[-1.9132,-1.9131]

-1.9132

[-1.9131,-1.9130]

-1.9130

[-1.9132,-1.9132]

-1.9132

GF

[0.5233,1.0884]

0.6352

[0.5233,0.5236]

0.5235

[0.5233,0.5233]

0.5233

DW

[-0.9985,-0.9362]

-0.9659

[-1.0000,-0.9963]

-0.9977

[-1.0000,-0.9898]

-0.9959

SP3

[0.0002,0.0050]

0.0011

[0.0002,0.0004]

0.0002

[0.0000,0.0004]

0.0000

AF4

[0.0069,0.0436]

0.0219

[0.0072,0.0532]

0.0132

[0.0000,0.0057]

0.0012

HF6

[-3.3209,-3.1921]

-3.2704

[-3.3211,-3.3211]

-3.3211

[-3.2634,-3.3219]

-3.3067

TR6

[-49.8360,-47.0466]

-48.5274

[-49.6214,-48.2239]

-48.9226

[-48.9561,-46.3729]

-47.5442

TR10

[-209.957,-200.714]

-206.159

[-208.322,-204.596]

-207.600

[-208.921,-206.164]

-207.984

GN10

[0.0074,0.0935]

0.0423

[0.0689,0.0764]

0.0727

[0.0000,0.0000]

0.0000

PF10

[0.0522,0.4194]

0.2008

[0.0622,38.8894]

11.3421

[0.0389,0.0705]

0.0563

WS12

[0.0000,0.0000]

0.0000

[0.0000,0.0000]

0.0000

[0.0000,0.0000]

0.0000

RF16

[0.7671,1.2721]

0.8270

[1.0722,1.4701]

1.1537

[0.0486,0.4363]

0.1416

TF20

[0.4656,1.4937]

0.9533

[0.0832,0.9522]

0.2133

[0.1462,0.3822]

0.3949

Point6: The motivation for the selection of a specific set of PSO parameters should be motivated.

Reply6: In our PSO algorithm, the value of w was 0.8; c1 = c2 =2. Parameters settings are reference by

Poli et al [1].

[1] Poli R ,  Kennedy J ,  Blackwell T . Particle swarm optimization[C]// IEEE Swarm Intelligence Symposium. IEEE, 2007.

Point7: The authors state that “Considering the errors in CFD calculations, the hull form results obtained by the two optimisation algorithms were basically the same”, the question is: why anyone should use the proposed version is not any significant improvement is demonstrated?

Reply: Consider two algorithms in optimizing space to find the best solution. So, the proposed version is not any significant improvement in value result. Through function example and CFD example, it can be found that optimization efficiency has improved.

Round 2

Reviewer 2 Report

The authors have fairly answered the previous comments and the paper has been improved and can be accepted in the present form.

Author Response

Thanks for your careful work and thoughtful suggestions that have helped improve this paper substantially.

Reviewer 3 Report

It is not clear if the domain is normalized as a unit hypercube or not, before running PSO. If yes, then what is the meaning of the dimensionality reduction? Usually the optimization algorithms seem to be more affected by the number of dimensions than by the domain extension.

For the analytical functions, the distance from the optimum should be used as a metric too (normalizing the distance, obviously).

What is the definition of "convergent algebra"?

The y-label of Figure 9 is missing.

Author Response

It is not clear if the domain is normalized as a unit hypercube or not, before running PSO. If yes, then what is the meaning of the dimensionality reduction? Usually the optimization algorithms seem to be more affected by the number of dimensions than by the domain extension.

Reply: The domain is not normalized as a unit hypercube, before running PSO. In the article, we only reduces the range of variables. Space reduction mainly includes two parts: size reduction (reduces the range of variables) and dimensionality reduction (reduces the number of variables). Range reduction is used to reduce the range of variables. Data mining is commonly used to analyse the internal relationships between parameters and performance to obtain a better optimised subspace where final optimisation is performed. Meanwhile, dimensionality reduction reduces the number of variables. Statistical methods such as sensitivity analysis and analysis of variance (ANOVA) are commonly used to analyse parameters so that parameters having less influence on the optimisation results can be eliminated.

For the analytical functions, the distance from the optimum should be used as a metric too (normalizing the distance, obviously).

Reply: Function example is single object optimisation problem. The distance from the optimum is the difference between the optimisation results and theoretical optimum, and it can be analyzed by fig. 9 in the paper.

What is the definition of "convergent algebra"?

Reply: Convergent algebra is rate of convergence. Rate of convergence is based on optimization time and simulation times. In this paper, we use the algebra as a convergence standard. The above discussion has been added in the revised version.

The y-label of Figure 9 is missing.

Reply: It has been revised and marked red in the paper.